# Do Boys Empathize Less than Girls? Exploring the Links Among Empathy, Gender and Sexist Attitudes in Adolescents

**DOI:** 10.3390/bs14111065

**Published:** 2024-11-07

**Authors:** Magaly Luisina García-Senlle, Manuel Martín-Fernández, Raquel Conchell, Sara Arrojo, Marisol Lila

**Affiliations:** 1Faculty of Psychology, Department of Social Psychology, University of València, 46010 Valencia, Spain; magaly.garcia@uv.es (M.L.G.-S.); manuel.martin@uv.es (M.M.-F.); 2Faculty of Philosophy and Education Sciences, Department of Comparative Education and History of Education, University of València, 46010 Valencia, Spain; raquel.conchell@uv.es; 3Center on Poverty and Community Development, Jack, Joseph and Morton Mandel School of Applied Social Sciences, Case Western Reserve University, Cleveland, OH 44106, USA; sara.arrojo@uv.es

**Keywords:** intimate partner violence, empathy, ambivalent sexism, adolescence

## Abstract

(1) Background: Intimate partner violence (IPV) remains a pervasive issue, particularly among adolescents. Its prevalence is still high despite intervention efforts, especially for younger generations. Empathy and sexism are factors linked with IPV that have shown gender differences. This study focuses on how gender moderates the association of empathy and sexist attitudes among adolescents in Spain; (2) Methods: A two-stage stratified cluster sampling method for collecting data from 516 adolescents (219 boys, 297 girls) aged 12–18 years was used. Participants completed the Ambivalent Sexism Inventory and the Interpersonal Reactivity Index. Hierarchical multiple regression analyzed the moderating role of gender in the relation between empathy and ambivalent sexism; (3) Results: Boys exhibited higher sexism levels than girls. Perspective taking negatively predicted hostile sexism. Empathic concern positively predicted hostile and benevolent sexism. Significant gender-specific patterns emerged: more empathic concern was linked with increased hostile sexism in boys; higher perspective taking was linked with increased benevolent sexism in boys, but was lower in girls; (4) Conclusions: There is a complex interplay among empathy, gender, and sexism, suggesting that IPV prevention programs should be tailored differently for boys and girls by considering broader socio-political contexts, addressing traditional gender norms, and promoting gender equality.

## 1. Introduction

Intimate partner violence (IPV) presents significant societal concerns, affecting mental health, social relationships, and economic stability at both individual and societal levels [1,2,3,4]. A key concern is intimate partner violence against women (IPVAW), a global public health crisis with intergenerational impacts on children exposed to violence, and a substantial economic burden due to medical expenses, lost productivity, and legal costs, resulting in staggering financial losses [2,5,6,7]. Given its pervasive impact, efforts have been made to understand what motivates this type of violence in an attempt to prevent it. A key factor in understanding IPVAW is the role of sexism, which emerges as a significant predictor of both perpetration and victimization [6,8,9]. Sexism, rooted in gender stereotypes and power differentials, encompasses both hostile and benevolent attitudes toward women [10]. Hostile sexism reflects overt hostility and derogation toward women, while benevolent sexism manifests as protective, yet patronizing attitudes that still reinforce gender inequality. Studies consistently find positive associations between both forms of sexism and IPVAW, with hostile sexism linked more to overt violence and benevolent sexism associated with controlling behaviors and psychological abuse [11,12,13,14].

In response to this concerning issue, numerous initiatives have emerged at various levels (i.e., global, national, and local) to combat IPVAW, some of them aiming to prevent and change sexist attitudes, among others [6,15,16]. Despite all these efforts, IPVAW prevalence rates remain alarmingly high, particularly among young generations. Globally, 26% of ever-partnered women and girls aged 15 years and older have experienced physical and/or sexual IPVAW in their lifetime [17,18]. Alarmingly, IPVAW onset occurs early, and it has high lifetime prevalence rates for adolescents aged 15–19 years (24%) [17]. In the European Union, the average lifetime prevalence of physical and/or sexual IPVAW stands at 22%, with significant rates for young adults aged 18–29 years [19,20]. Notably, recent research highlights higher IPVAW prevalence rates for younger age groups in Spain when compared to older age groups, the country in which this study was conducted, which underscores the need for a nuanced exploration of the socio-cultural factors that influence these trends [21].

These startling data call for a closer look at the factors that maintain the presence of this type of violence in our society, particularly in the younger generations. IPVAW represents a multifaceted phenomenon that is influenced by various factors across different social ecology levels, in which empathy emerges as a crucial individual factor linked with intimate violence dynamics [6,8,9,22,23]. Empathy, regarded as a fundamental aspect of human interaction, has been consistently related to prosocial behavior and attitudes in both its affective and cognitive or perspective-taking dimensions [24,25,26]. As stated by Le Brun et al. [27], empathy is vital in addressing IPV, influencing how victims are perceived and supported after disclosing their experiences. By validating victims’ emotions, empathy fosters positive social reactions, encourages help-seeking behavior, and challenges harmful stereotypes. In addition to this, empathy also plays a complex role related to IPV perpetration. Ulloa and Hammett’s [28] dyadic study found that men with lower empathy levels are not only more prone to perpetrate IPV but are also more likely to be victims of it. Similarly, women whose male partners exhibit lower empathy levels are more inclined to perpetrate IPV and be victimized by it, which highlights the intricate relation between empathy and IPV dynamics. Moreover, Romero-Martínez et al. [29,30] observe how IPVAW offenders with higher alexithymia (i.e., difficulty identifying and expressing emotions) levels, as well as antisocial and borderline personality traits, exhibit severer impairments in empathy and are at a higher risk of perpetrating this type of violence.

Understanding the interplay between empathy and sexism in shaping attitudes and behaviors related to IPV is essential for developing targeted interventions and promoting gender equality and violence efficacious prevention. Indeed, several studies demonstrate the existing relation between sexism and empathy, with lower empathy levels being associated with more sexist beliefs [31,32,33,34,35,36]. In this sense, cognitive empathy may be particularly relevant in mitigating hostile sexist beliefs, as perspective taking could foster more egalitarian attitudes, while affective empathy, though generally associated with prosocial outcomes, may have a complex relationship with benevolent sexism, as some forms of protective attitudes might be driven by genuine concern [32,33,36].

In addition to the direct link between empathy and sexism, gender differences must be considered, as some authors report that this connection may be mediated by them [27,32,35]. Several findings seem to point at a tendency where women exhibit more empathy and hold fewer sexist attitudes than men [12,27,32,36]. Regarding the mediating role of gender in the relationship between empathy and sexism, Garaigordobil’s review [32] shows that the link between empathy and sexism is stronger in men. When analyzing this gendered relationship in depth, Villanueva-Blasco et al. [36] found that sexism significantly predicts both perpetration and victimization of teen dating violence, with empathy serving as a crucial mediating factor influenced by gender. In this regard, they observed that boys are more likely to hold negative attitudes towards women (i.e., hostile sexism), which can lead to aggressive behaviors in relationships. For boys, feelings of personal distress, a dimension of affective empathy, can increase the likelihood of acting on these sexist beliefs in harmful ways. On the other hand, girls generally have higher levels of empathy, which helps them understand and connect with others emotionally. This empathy, along with their ability to communicate assertively, can reduce the chances of them engaging in violent behavior, even if they hold some sexist beliefs.

Gender differences in these dynamics underscore the imperative for customized interventions, particularly in light of reports that indicate the growing acceptance of sexist and traditional gender norms, as well as tolerance toward IPV, by young males compared to their female counterparts [37,38]. This phenomenon is particularly relevant within a societal context where some men may perceive advancements in gender equality as a threat to traditional power structures, which potentially elicits reactionary responses [21,39,40,41,42]. Van Laar et al. [41] illustrate the diverse perspectives that men hold of gender equality progress. While some perceive it as advantageous, others resist or express uncertainty. These attitudes are shaped by factors like fearing loss of social status, adhering to traditional masculinity norms and perceiving women’s advancements as threatening. This resistance or uncertainty frequently arises from the belief that women’s progress can potentially put men at a disadvantage, which could explain gender differences found regarding sexism.

This divergence in attitudes may contribute to the persistence of IPV, particularly among younger generations. Of these concerning trends in IPV prevalence and the multifaceted dynamics that involve empathy and sexism, considering how these factors are manifested among younger generations is paramount. Adolescence represents a crucial developmental stage characterized by significant transitions and identity formation, in which attitudes and behaviors regarding gender roles and relationships are often solidified [18,43,44]. As highlighted by Gracia et al. [21], the high IPV prevalence among youths necessitates re-evaluating current prevention strategies. Examining how both genders engage with sexist beliefs can inform the adaptation and enhancement of interventions tailored to younger generations’ needs. Moreover, understanding empathy dynamics becomes necessary within the context of a potential backlash effect, where boys may perceive feminist demands or gender equality as threatening [39,40,41]. Existing studies suggest that girls have higher empathy levels than boys [45], but contemplating how the current social context may be affecting boys’ empathy is crucial. Understanding the interplay between sexist beliefs and empathy is necessary for determining the importance of empathy in prevention efforts and whether interventions should be tailored differently for boys and girls based on their perceptions and empathetic dynamics. Therefore, integrating an understanding of empathy differences between genders into prevention efforts is essential for not only effectively addressing the root causes of IPV but also for promoting healthy relationship dynamics among adolescents.

### The Present Study

Given the prevalence and detrimental consequences of IPV, particularly among young populations [5,17], it is essential to revisit key variables such as empathy and gender to better inform prevention strategies tailored to adolescents’ specific needs. Empathy emerges as a crucial factor that influences attitudes and behaviors related to IPV, with lower empathy levels associated with increased perpetration, victimization, and sexist attitudes [27,28,29,30,36]. Recent findings in adolescents suggest that empathy’s influence on sexist attitudes is mediated by gender, with boys’ higher personal distress linked to acting on hostile sexist beliefs and girls’ greater empathy reducing violent behavior despite some sexist attitudes [36].

This study aimed to investigate how empathy and gender interact to predict sexism among adolescents in a Spanish sample, building on current research within the context of IPV. We hypothesized that higher empathy would be associated with lower sexism, but this relationship would be more pronounced for girls, as their empathy would be more likely to mitigate sexist attitudes compared to that of boys. The present study aimed to extend the knowledge on the interaction of empathy and gender in predicting sexism among adolescents. Building on previous research that highlights the complex interplay among empathy, gender, and attitudes toward sexism through young generations [32,36,38,45], we sought to broaden our understanding of these dynamics in the IPV prevention context.

## 2. Materials and Methods

### 2.1. Participants

A two-stage stratified cluster sampling method was used to collect the dataset. Initially, a list of all the public and private secondary schools in the Valencian community (Spain) was drawn up. Subsequently, three public and three private schools were selected. In the second phase, five groups from each school were randomly selected to participate in this study, which resulted in 516 responses. The sample consisted of 219 boys and 297 girls, whose age range was between 12 and 18 years (*M* = 15.31 years, *SD* = 1.52). Most participants were of Spanish nationality (89.8%), and 10.2% identified as having other nationalities. Their current levels of education were as follows: 19.8% in grade 2 of Compulsory Secondary Education (CSE); 24.8% in grade 3 of CSE; 17.1% in grade 4 of CSE; 26% Baccalaureate; and 12.3% Vocational and Technical Training.

### 2.2. Instruments

This study applied the Ambivalent Sexism Inventory (ASI) [10], which is adapted for use with adolescents in Spanish [46]. This inventory consists of two subscales: hostile and benevolent sexism. Each one comprises 10 items rated on a 6-point Likert scale: 1 = disagree strongly, 6 = agree strongly. Hostile sexism concerns biased views and discriminatory attitudes toward girls, based on the perception of inferiority and submissiveness to boys (e.g., “Boys should control who their girlfriends associate with”). In contrast, benevolent sexism reflects a positive, albeit stereotypical, representation of girls in specific roles (e.g., “Girls are more sensitive to the feelings of others than boys”). The ASI has undergone adaptation and validation in various socio-cultural contexts [47,48,49,50,51], including adolescent populations [47,52]. The adolescent version of the scale has been associated with traditional sexist ideologies [47] and perceptions of severity of adolescent domestic violence [53,54]. Strong internal consistency was evidenced for the present sample for both dimensions: hostile sexism α = 0.89; benevolent sexism α = 0.84.

The short form of the Interpersonal Reactivity Index (B-IRI) [24,55] was used to measure empathy. This instrument encompasses 16 items that evaluate cognitive and emotional empathic responses on a 5-point scale: 1 = does not describe me well; 5 = describes me very well. Cognitive empathy is assessed with the perspective taking (e.g., “I try to look at everybody’s side of a disagreement before I make a decision”) and fantasy (e.g., “When I watch a good movie, I can very easily put myself in the place of a leading character) subscales, while emotional empathy is measured by the empathic concern (e.g., “I often have tender, concerned feelings for people less fortunate than me”) and personal distress (e.g., “In emergency situations, I feel apprehensive and ill-at-ease”) subscales. The empathic concern and perceived distress dimensions have been correlated positively with emotional fragility and femininity, and negatively with other relevant constructs like masculinity. Perspective taking and fantasy are not related to masculinity, but are positively associated with emotional fragility [55]. The reliability of the scale in the original study, assessed using Cronbach’s alpha, ranged from 0.68 to 0.79 for the four dimensions, which indicates adequate internal consistency. For this study, internal consistency was also adequate and ranged from 0.65 to 0.83.

### 2.3. Procedure

Before distributing the survey, meetings were held with teachers at each high school. Participants received information about this study’s objectives and provided anonymous responses. The survey administered to participants included the ASI and B-IRI scales, along with a series of socio-demographic questions covering age, gender, nationality, high school name, current grade level, socio-economic status, and relationship status. Participants were instructed to complete all items of the scales, and researchers were available during the administration of the instruments in case some of the participants had any doubts. Hence, there were no missing values. Data collection occurred from November 2021 to May 2022 through the in-person administration of questionnaires. Approval for this study was obtained from the Conselleria de Educación, Cultura y Deporte (Generalitat Valenciana, Spain).

### 2.4. Analysis

A descriptive analysis was first carried out for girls and boys separately. The raw scores of all measures were computed and the mean, standard deviation, maximum, minimum, skew, and kurtosis statistics were obtained for both hostile and benevolent sexism and for the four subscales of the B-IRI (i.e., perspective taking, empathic concern, perceived distress, and fantasy).

In order to investigate the potential moderating role of gender in the relationship between empathy and sexist attitudes, a moderation analysis was conducted using hierarchical multiple regression in IBM SPSS Statistics, version 27. The analysis consisted of three steps and followed the recommendations outlined by Hayes and Rockwood [56]. In the first step, gender, which was coded as 0 = girl or 1 = boy, was entered as a predictor variable. Age was considered a potential covariate and was entered into the regression model. In the second step, the empathy scores were entered as a predictor variable. In the third step, the interaction term between gender and empathy was entered to examine whether gender moderated the relationship between empathy and sexist attitudes. The process was repeated for all the sexism dimensions (i.e., hostile and benevolent).

To ensure a statistical power above 0.80 for this analysis, a sample of at least 97 participants was needed. In our study, the sample size was 516, which was more than enough to meet this requirement. Before conducting the moderation analysis, assumptions were assessed to ensure the robustness of the results. Specifically, multicollinearity diagnostics were examined to ensure that the predictor variables did not highly correlate. Residual plots were inspected to verify that the assumptions of linearity, homoscedasticity, and normality of residuals were met.

To interpret the moderation effect, significance levels were determined using an alpha criterion of *p* < 0.05. If the interaction term was statistically significant, simple slope analyses were run to examine the nature of the moderation effect.

## 3. Results

### 3.1. Descriptive Analysis

When the descriptive statistics were computed for girls and boys, we observed that, on average, girls and boys presented low levels of ambivalent sexism (the category where the agreement responses of the items begin) (see Table 1), with mean scores below 4. Girls tended to show lower levels than boys in both hostile and benevolent sexism (i.e., 2.06 and 2.57, respectively), implying that they tended to show disagreement with most of the items. Boys also showed on average disagreement with the items of the ASI, but presented slightly higher means in both variables than girls (i.e., 3.09 and 3.17 for hostile and benevolent sexism).

Regarding empathy, we found that in general boys and girls presented some variability in their responses to B-IRI, with average responses around the central category (i.e., 3 = not agree nor disagree). Boys showed, however, smaller mean scores in all subscales of the B-IRI (see Table 1). These differences were not too prominent for empathic concern (around 0.20), and perspective taking and perceived distress (around 0.50), implying that in general, both genders present similar scores in these variables. In the fantasy subscale, the difference was higher, suggesting that girls tended to empathize on average more with fictional characters than boys in our sample.

### 3.2. Hostile Sexism

Hierarchical multiple regression was conducted to explore the moderation effect of gender on the relationship between empathy and hostile sexism. In the first model, gender and age were entered as independent predictors of hostile sexism. This model explained 25% of total variance (adjusted R^2^ = 0.25, *F*(*df*) = 85.76(2), *p* < 0.001). Gender (β = 0.46, *t* = 19.47, *p* < 0.001) and age (β = −0.13, *t* = −5.94, *p* < 0.001) predicted hostile sexism, with boys having more hostile attitudes than girls, and with older participants obtaining lower hostile sexism scores (see Table 2).

In the second step, the four empathy dimensions were introduced. This model accounted for 27% of the total variance of hostile sexism (adjusted R^2^ = 0.27, *F*(*df*) = 33.03(6), *p* < 0.001). As can be seen in Table 2, in this model, gender and age still predicted hostile sexism in the same direction (β = 0.47, *t* = 18.87, *p* < 0.001; β = −0.13, *t* = −5.60, *p* < 0.001, respectively). For empathy, empathic concern and perceived distress were positive predictors of the dependent variable, and those respondents with higher scores on these dimensions took more hostile sexist attitudes (β = 0.15, *t* = 3.17, *p* = 0.002; β = 0.12, *t* = 2.70, *p* = 0.007, respectively). Perspective taking was shown to be a significant negative predictor of hostile sexism, and scoring higher for this variable was linked with a lower hostile sexism score (β = −0.13, *t* = −2.76, *p* = 0.002).

Finally, when adding the interaction between gender and all the empathy dimensions, the percentage of explained variance increased up to 30% (adjusted R^2^ = 0.30, *F*(*df*) = 23.15(10), *p* < 0.001). This model significantly increased the variance accounted for by 3.4% (∆R^2^ = 0.034, *p* < 0.001). When introducing the gender–empathy interaction, gender and empathic concern lost their predictive ability (see Table 2), with the moderator role of gender only being significant for the empathic concern dimension. For boys, hostile sexism increased 0.84 units for each standard deviation increase in empathic concern (β = 0.84, *t* = 4.25, *p* < 0.001), but there was no such effect for girls. In particular, boys showing higher scores of empathic concern also presented higher scores for hostile sexism, whereas empathic concern had no effect on girls’ scores for this type of sexism (see Figure 1). Age remained constant as a covariate predictor of hostile sexism, with lower hostile sexism rates with increasing age (β = −0.13, *t* = −5.56, *p* < 0.001).

### 3.3. Benevolent Sexism

Further hierarchical multiple regression was conducted to assess whether the relation between empathy and benevolent sexism was moderated by gender (see Table 3). The first step introduced gender and age as the sole predictors of benevolent sexism, with this model accounting for 11% of total variance (adjusted *R*^2^ = 0.11, *F*(*df*) = 33.21(2), *p* < 0.001). Age predicted benevolent sexism, and older participants had lower benevolent sexism levels than younger ones (β = −0.15, *t* = −3.53, *p* < 0.001). Regarding gender, boys had, on average, higher benevolent sexism levels than girls (β = 0.28, *t* = 6.54, *p* < 0.001).

When the four empathy dimensions were entered in the second step, the model explained 13% of the total variance of benevolent sexism (adjusted *R*^2^ = 0.13, *F*(*df*) = 14.17(6), *p* < 0.001). Gender and age continued to predict benevolent sexism in the same direction (β = 29, *t* = 6.52, *p* < 0.001; β = −0.16, *t* = −3.69, *p* < 0.001). As can be seen in Table 2, only empathic concern significantly predicted an increase in benevolent sexism in this second stage (β = 0.17, *t* = 3.41, *p* = 0.001).

The third hierarchical regression step consisted in considering the moderator effect of gender on each empathy dimension. This final model explained 2.1% more of the total variance of benevolent sexism than the previous step (change *R*^2^ = 0.021, *p* = 0.013). In particular, the percentage of variance explained in this model increased up to 15% (adjusted *R*^2^ = 0.15, *F*(*df*) = 9.92(10), *p* < 0.001). Gender and empathic concern were not significant predictors in this model, but perspective taking became significant for predicting benevolent sexism (β = −0.17, *t* = −2.60, *p* = 0.010). The gender–perspective taking interaction was also a significant predictor (β = 0.38, *t* = 2.05, *p* = 0.041). Taking both the main and the interaction effect together, we found for boys that benevolent sexism increased by 0.21 units for each standard deviation increase in perspective taking. For girls, benevolent sexism decreased by 0.17 units for each standard deviation increase in perspective taking (see Figure 2). In this model, age continued to be a significant covariate for predicting benevolent sexism in the same direction (β = −0.15, *t* = −3.68, *p* < 0.001).

## 4. Discussion

The main goal of this study was to investigate how empathy and sexism interact, and to examine any gender differences in the adolescent population. As adolescence represents a critical developmental stage for establishing healthy intimate relationship patterns [18,43,44], intervention programs that target IPV prevention in this age group would greatly benefit from identifying and addressing the pertinent factors that impact this population and how. Moreover, it is essential to understand whether interventions should be tailored differently for boys and girls based on their specific needs [57,58]. Empathy, linked with a predisposition to show prosocial behavior [25,26], has emerged as a protective factor against IPV [27,28,29,30,36]. Conversely, sexism, which is often inversely related to empathy [31,32,33,34,35,36], has been robustly linked with a higher IPV risk [11,12,13,14], and higher sexism levels have been correlated with higher odds of perpetration and victimization. Moreover, both empathy and sexism exhibit gender-specific patterns, with studies indicating variations between males and females in both their levels and manifestations [32,36,45].

Our study revealed that boys exhibited higher hostile and benevolent sexism levels than girls, which is consistent with former research indicating that younger individuals, particularly males, are more prone to endorse sexist attitudes [32,36,59]. Furthermore, we found that older adolescents tended to exhibit lower hostile and benevolent sexism levels, and this relationship remained constant regardless of the interaction effect of empathy and gender. These findings suggest potential developmental shifts in attitudes toward gender. With age, although its relationship with sexism and other attitudes linked with IPV is unclear, some authors suggest a U-shaped pattern in which age can be inversely linked with them, and some argue that this relationship might be mediated by the level of education, which usually increases with age [8,59,60,61,62]. In empathy terms, perspective taking was a negative predictor of hostile sexism, which indicates that the ability to understand others’ perspectives may mitigate hostile sexist attitudes. Conversely, empathic concern and perceived distress were positive predictors of hostile sexism. For benevolent sexism, empathic concern was a positive predictor, which suggests that higher empathic concern levels can be associated with increased benevolent sexist attitudes. This nuanced relationship between the different empathy dimensions and sexism types highlights the complexity of these constructs and their interactions.

In fact, when accounting for the gender–empathy interaction for both hostile and benevolent sexism, gender-specific patterns emerged to explain how the empathy levels of both girls and boys related to sexism. The gender–empathetic concern interaction was significant for hostile sexism. Specifically for boys, more empathic concern was associated with increased hostile sexism, while this relationship was not observed for girls. This suggests a paradoxical effect where increased empathy in boys might not translate into fewer sexist attitudes, potentially due to misinterpretation or misapplication of empathic feelings. The literature consistently shows that people tend to exhibit more empathy for others when they belong to their own social group [63,64,65]. In this case, one might ask to whom do boys direct that empathic concern? Boys may possibly empathize more with other males in the presented sexist scenarios, with this misdirected empathy potentially supporting the maintenance of in-group biases and out-of-group hostility, which would further entrench gendered prejudices and behaviors. Understanding this possible bias necessitates acknowledging the social context in which male adolescents operate, which may significantly impact their interpretations and expressions of empathy. The rise in political polarization, the resurgence of right-wing ideologies, and increasing gender equality may contribute to polarized attitudes and resistance to changing norms [21,39,66,67,68]. These societal shifts could lead boys to perceive feminist advancements as threats, which would further entrench sexist attitudes. Zero-sum beliefs posit that gains in gender equality for women are perceived as losses for men and perceived competition between genders, which imply men’s resistance and hostile sexism [39,40,41,42,69,70]. This would align with the fact that hostile sexism operates through envious and resentful prejudices, often by responding to perceived threats to male power and sexuality [10,12]. In Spain, recent reports indicate significant gender differences among individuals aged 15–29 years for feminism support and the perception of gender-based violence as a serious social problem, with decreases reported for boys and increases for girls [37]. Understanding these biases and contextual influences is essential for comprehending how gender and empathy interact to predict hostile sexism among adolescents.

For benevolent sexism, the gender–perspective taking interaction was also significant. The boys with higher perspective taking showed increased benevolent sexism, whereas the girls with higher perspective taking displayed decreased benevolent sexism. What this indicates is that although perspective taking fosters egalitarian attitudes in girls, it might reinforce protective and patronizing attitudes in boys. This finding is consistent with research that suggests that benevolent sexism is often rooted in protective and paternalistic feelings that can be enhanced by empathic understanding [12]. One possible explanation for this gender-specific pattern is that boys’ empathic concern may translate into a form of empathy that aligns with traditional gender roles, where they see themselves as protectors of women [71,72,73,74]. Although this protective stance is seemingly positive, it aligns with principles of benevolent sexism, which frames women as needing men’s care and protection [10,12]. Thus, boys with higher perspective taking may be more likely to endorse benevolent sexism because they might perceive their empathy as a justification for safeguarding women, which is reinforcing traditional and patronizing gender roles. Some studies have also shown that benevolent sexism can be driven by the need for security. For example, women’s endorsement of benevolent sexism increases when they believe that men take more hostile attitudes toward them and when their fear of crime is heightened [59]. This is particularly relevant in the current social context, where heightened awareness of IPV and other forms of gender-based violence are considered a serious social problem and the need to intervene is perceived. This perception is reinforced by various legal measures designed to protect women [17,75,76,77]. As Castro et al. [76] argue, some unexpected consequences may derive from macrosocial legislative and political changes. One of them, already noted, may be linked with the possible backlash effect that is on the increase in countries where gender equality is at its highest (see ‘The Nordic Paradox’, [69]). The results of this study pose the question of whether the increase in awareness and legal measures to ensure gender equality and to prevent IPV may have inadvertently positioned women in a role of ‘vulnerable’, where women need protection. In this light, boys may perceive empathic concern as a protective mechanism by responding to societal cues about women’s vulnerability and by adopting benevolent sexist attitudes. Consequently, empathic concern in boys may inadvertently sustain traditional gender roles and benevolent sexism.

The findings of this study offer several practical implications for developing effective interventions to reduce IPV and sexism among adolescents. Given the observed gender-specific patterns in empathy and sexism, it may be crucial to design and implement IPV prevention programs that are tailored differently for boys and girls. Although different IPV prevention programs have proven effective [78,79,80], some evidence suggests a research gap and that robust well-established measures that specifically address boys are lacking [81,82]. Intervention programs must consider the broader socio-political contexts that influence adolescents’ attitudes toward gender and sexism, particularly for boys. While effective programs typically address gender norms, it is important to extend these programs to include topics that address boys’ potential feelings of threat. Specifically addressing traditional masculinity norms can help boys to not only reconcile their roles within the gender equality framework, but to also take fewer paternalistic attitudes, as long as it is approached from a gender-conciliatory pedagogical framework that does not promote polarization and, in this case, the blaming of men as a collective [83,84,85]. As Stewart et al. [82] argue, it is essential to direct these intervention efforts to boys in making them participants in change, and not just allies, by them understanding that this is something that also affects them, even in their own gender, and not just as opposed to girls.

Despite this study’s findings, it has several limitations that should be acknowledged. First, although the sample size was adequate, its findings are specific to the Spanish adolescent population and may not be generalizable to other cultural contexts. Future research should include diverse populations to validate these findings across different cultural settings. Additionally, this study relied on self-reported measures of empathy and sexism, which can be subject to social desirability bias. Future studies should consider incorporating objective measures or multi-informant approaches to enhance data reliability. Moreover, the cross-sectional nature of this study prevents causal relationships from being established. Longitudinal studies are needed to explore how empathy and sexist attitudes evolve over time and their long-term impact on IPV perpetration and victimization. Moving forward, future research should also explore additional factors that influence the relationships among empathy, sexism, and IPV, such as cultural and societal norms. Intervention studies are also required to evaluate the effectiveness of targeted interventions in reducing IPV perpetration and victimization among adolescents.

In conclusion, the present study contributes to a deeper understanding of the complex interplay among empathy, gender, and sexism in shaping adolescent attitudes toward IPV. These findings highlight the need to continue exploring novel avenues for intervention and prevention efforts that aim to foster gender equity and lower IPV prevalence in adolescent and young populations. By addressing the underlying socio-psychological factors that drive sexist beliefs, we can pave the way toward healthier, more equitable relationships and communities.

## Figures and Tables

**Figure 1 behavsci-14-01065-f001:**
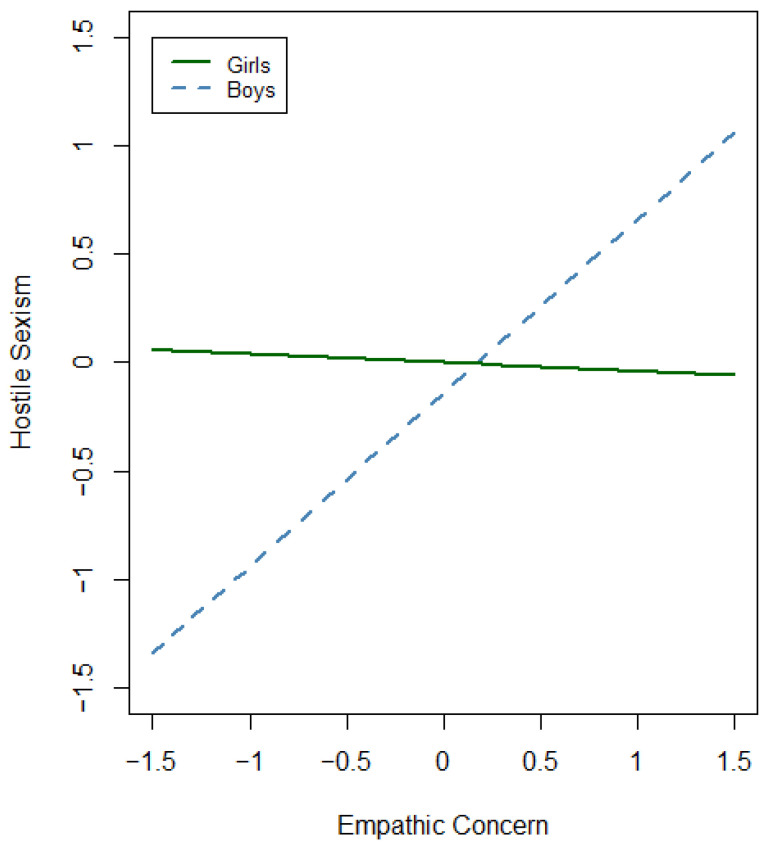
Moderation effect of gender in empathic concern for hostile sexism.

**Figure 2 behavsci-14-01065-f002:**
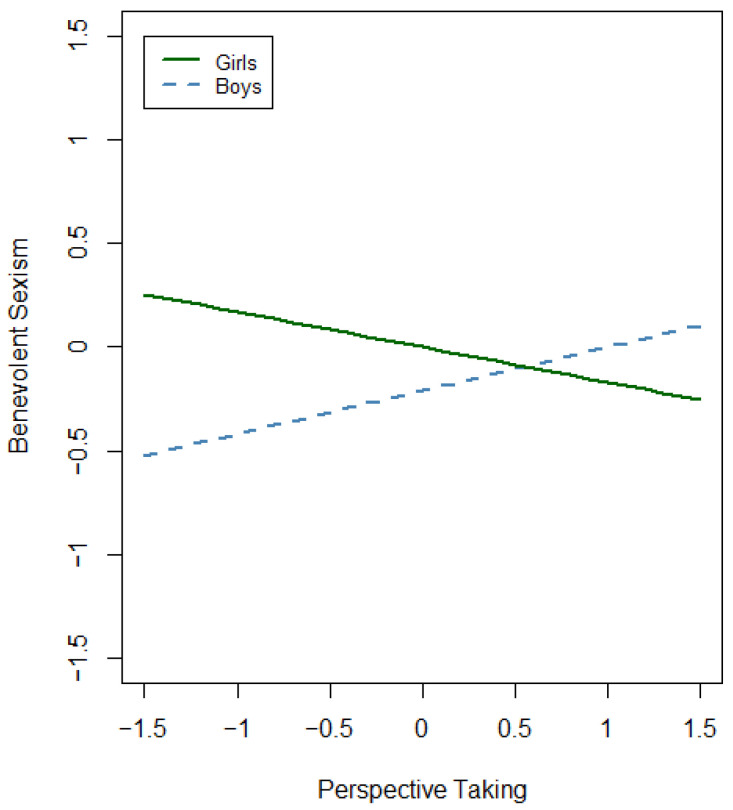
Moderation effect of gender in perspective taking for benevolent sexism.

**Table 1 behavsci-14-01065-t001:** Descriptive statistics for girls and boys.

Gender	Variable	M	SD	Min	Max	Skew (s.e.)	Kurtosis (s.e.)
Girls	Hostile sexism	2.06	0.83	1.00	5.20	1.01 (0.14)	0.95 (0.28)
	Benevolent sexism	2.57	0.95	1.00	5.60	0.50 (0.14)	−0.05 (0.28)
	Perspective taking	2.63	0.91	1.00	5.00	−0.04 (0.14)	−0.39 (0.28)
	Empathic concern	3.77	0.74	1.75	5.00	−0.07 (0.14)	−0.55 (0.28)
	Perceived distress	3.29	0.88	1.00	5.00	0.31 (0.14)	−0.67 (0.28)
	Fantasy	3.10	1.13	1.00	5.00	−0.38 (0.14)	−0.91 (0.28)
Boys	Hostile sexism	3.09	1.05	1.00	5.80	0.14 (0.16)	−0.75 (0.33)
	Benevolent sexism	3.17	0.92	1.00	5.50	−0.31 (0.16)	−0.28 (0.33)
	Perspective taking	2.11	0.88	1.00	5.00	0.69 (0.16)	0.75 (0.33)
	Empathic concern	3.59	0.75	1.25	5.00	0.17 (0.16)	−0.23 (0.33)
	Perceived distress	2.88	0.90	1.00	5.00	1.00 (0.16)	−0.49 (0.33)
	Fantasy	2.29	1.06	1.00	5.00	−0.34 (0.16)	−0.18 (0.33)

Note. M = mean, SD = standard deviation, Max = maximum, Min = minimum, s.e. = standard error.

**Table 2 behavsci-14-01065-t002:** Hierarchical regression model for hostile sexism.

Model	Variable	*β*
Socio-demographics	Gender	0.46 ***
	Age	−0.13 **
Main empathy effects	Gender	0.47 ***
	Age	−0.13 **
	Empathy	
	Perspective taking	−0.13 **
	Empathic concern	0.15 **
	Perceived distress	0.12 **
	Fantasy	−0.02
Moderation effects	Gender	−0.14
	Age	−0.13 **
	Empathy	
	Perspective taking	−0.01
	Empathic concern	−0.04
	Perceived distress	0.08
	Fantasy	0.05
	Interactions	
	Gender × Perspective taking	−0.18
	Gender × Empathic concern	0.84 ***
	Gender × Perceived distress	0.10
	Gender × Fantasy	−0.12

Note. *β* = standardized coefficient. ** *p* < 0.01; *** *p* < 0.001.

**Table 3 behavsci-14-01065-t003:** Hierarchical regression model for benevolent sexism.

Model	Variable	*β*
Socio-demographics	Gender	0.277 ***
	Age	−0.149 ***
Main empathy effects	Gender	0.295 ***
	Age	−0.156 ***
	Empathy	
	Perspective taking	−0.086
	Empathic concern	0.172
	Perceived distress	0.057
	Fantasy	0.019
Moderation effects	Gender	−0.208
	Age	−0.154 ***
	Empathy	
	Perspective taking	−0.170 *
	Empathic concern	0.092
	Perceived distress	0.072
	Fantasy	0.055
	Interactions	
	Gender × Perspective taking	0.378 *
	Gender × Empathic concern	0.318
	Gender × Perceived distress	−0.066
	Gender × Fantasy	−0.113

Note. *β* = standardized coefficient. * *p* < 0.05; *** *p* < 0.001.

## Data Availability

The data presented in this study are available on request from the corresponding author due to privacy and ethical restrictions.

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
