# Peer review of "Do Boys Empathize Less than Girls? Exploring the Links Among Empathy, Gender and Sexist Attitudes in Adolescents"

_behavsci, 2024, doi:10.3390/bs14111065_

Round 1
Reviewer 1 Report
Comments and Suggestions for Authors
Thank you for the opportunity to review this manuscript. This manuscript describes an important and timely question of how dimensions of empathy are linked to ambivalent sexist beliefs (which are foundational beliefs for IPVAW) as well as whether these links are mediated by gender. The paper found evidence to suggest that dimensions of empathy are indeed differentially linked to sexism based on gender. Overall, the paper provides new and important findings that could contribute to the field of sexism and IPVAW. However, I noted some suggestions particularly regarding the framing and theory guiding the presentation of the study that could strengthen the understanding and impact of this study’s contributions. I describe these below.
INTRODUCTION
The introduction opens (and continues throughout) by highlighting the concerns related to IPVAW prevalence and describing the key measures of interest as foundational for the justification and prevalence of IPVAW -- which is indeed important and compelling. However, this framing of the introduction resulted in an expectation that specific attitudes related to IPVAW (such as violence myths) or IPVAW perpetration/victimization would be tested in the present study. Revising the framing to more clearly motivate the tested measures would improve the reading experience.
In contrast, the theoretical background of ambivalent sexism was briefer relative to the opening sections on IPVAW. Because this is the key measure of interest in the present study, more elaboration on ambivalent sexism theory would be beneficial and more elaboration on ambivalent sexism as it would relate to IPVAW would also serve to address the concern listed above.
The introduction would also benefit from more background and theory related to the key predictor of empathy:
-Firstly, the sections describing empathy in previous literature is limited to describing prior findings, while the “why” behind these previous patterns/driving the theoretical links tested in this study are not explicitly stated. What mechanisms specifically are proposed to link empathy and sexist beliefs, and what mechanisms would result in differences between girls and boys?
-Relatedly, the introduction mentions the backlash effect which is later used to explain the results found between more empathic concern and more hostile sexism (which is an interesting and compelling take-away indeed!), but prior to that, the reader is left unclear with the contribution/connection of that introduced concept in the introduction.
-Similarly, the study tests empathy multidimensionally, and finds differential links for these dimensions, but the introduction does not delve into the different dimensions of empathy and how/why they might be similarly or differently linked to ambivalent sexism.
-The points mentioned above also results in a lack of clear hypotheses in the study. Any hypotheses should be more clearly motivated and explicitly stated.
Thus, more explicitly addressing the proposed mechanisms or theoretical links between the specific measures of interest (both ambivalent sexism and multidimensional empathy) would benefit the framing of the study and strengthen its theoretical contributions for the field.
Additional suggestions:
-There were some instances in the introduction (e.g., line 48, line 72) where previous research described instances where groups were “higher” in some factor but to whom this comparison was being drawn was unclear for the reader. For instance, on line 48, were prevalence rates higher in Spain relative to other nations? Or compared to previous prevalence rates in Spain?
-In the Method: Analysis section, it is mentioned that age was entered into the model if deemed necessary. Because it was entered into both models the addition “if deemed necessary” is not needed and a bit confusing on initial read.
-In the results, include the nonsignificant simple effect beta predicting hostile sexism for girls.
-Provide some descriptive information (Ms and SDs) of the key measures of interest (overall or by gender) to provide more context about the findings, especially since group differences by gender and age is described in the results. What were the mean differences? Was empathy endorsement overall high? Or moderate? etc.
Author Response
Reviewer 1
Comment
The introduction opens (and continues throughout) by highlighting the concerns related to IPVAW prevalence and describing the key measures of interest as foundational for the justification and prevalence of IPVAW -- which is indeed important and compelling. However, this framing of the introduction resulted in an expectation that specific attitudes related to IPVAW (such as violence myths) or IPVAW perpetration/victimization would be tested in the present study. Revising the framing to more clearly motivate the tested measures would improve the reading experience.
In contrast, the theoretical background of ambivalent sexism was briefer relative to the opening sections on IPVAW. Because this is the key measure of interest in the present study, more elaboration on ambivalent sexism theory would be beneficial and more elaboration on ambivalent sexism as it would relate to IPVAW would also serve to address the concern listed above.
Response
Thank you for your appreciation. We have addressed this issue by introducing sexism earlier in the introduction and adding to the explanation on ambivalent sexism theory (lines 35-44).
Comment
The introduction would also benefit from more background and theory related to the key predictor of empathy:
-Firstly, the sections describing empathy in previous literature is limited to describing prior findings, while the “why” behind these previous patterns/driving the theoretical links tested in this study are not explicitly stated. What mechanisms specifically are proposed to link empathy and sexist beliefs, and what mechanisms would result in differences between girls and boys?
-Similarly, the study tests empathy multidimensionally, and finds differential links for these dimensions, but the introduction does not delve into the different dimensions of empathy and how/why they might be similarly or differently linked to ambivalent sexism.
Response
Thank you for your suggestions on explaining the concept of empathy. We have addressed this issue expanding the mechanisms through which empathy may be working (lines 64-68 and 82-86). Also, we have mentioned the two empathy dimensions defined by Davis (line 64).
Comment
Relatedly, the introduction mentions the backlash effect which is later used to explain the results found between more empathic concern and more hostile sexism (which is an interesting and compelling take-away indeed!), but prior to that, the reader is left unclear with the contribution/connection of that introduced concept in the introduction.
Response
Thank you for noticing this. To address it, we have reformulated the way in which it is expressed so it is clear that in the introduction section its importance relies on being a contextual and sociological framework (e.g., line 106) and how it may be affecting boys and girls differently (line 114).
Comments
The points mentioned above also results in a lack of clear hypotheses in the study. Any hypotheses should be more clearly motivated and explicitly stated.
Response
Thank you for your comment. Indeed, the study had an exploratory nature that we hope now is clearer by suggesting some hypothesis based on recent literature also added to the theoretical framework (lines 148-150).
Comment
There were some instances in the introduction (e.g., line 48, line 72) where previous research described instances where groups were “higher” in some factor but to whom this comparison was being drawn was unclear for the reader. For instance, on line 48, were prevalence rates higher in Spain relative to other nations? Or compared to previous prevalence rates in Spain?
Response
Thank you for highlighting this issue we have addressed by specifying the comparison group to which the expression is referring to (line 55).
Comment
In the Method: Analysis section, it is mentioned that age was entered into the model if deemed necessary. Because it was entered into both models the addition “if deemed necessary” is not needed and a bit confusing on initial read.
Response
Thank you for pointing this out. It is true that it was deemed necessary in both cases, and hence we have rephrased this sentence to avoid misunderstandings (line 217).
Comment
In the results, include the nonsignificant simple effect beta predicting hostile sexism for girls.
Response
The beta for boys and girls is the same and was already introduced in both models. Since gender was coded as a dummy variable (0=girls, 1=boys), the value of this beta reflects the change in the model intercept when gender is taken into account. To better illustrate the moderation effect, and following reviewer 2 suggestion also on this same issue, we have included a plot depicting the interaction found in our results (pages 7 and 9). We hope it is clearer now.
Comment
Provide some descriptive information (Ms and SDs) of the key measures of interest (overall or by gender) to provide more context about the findings, especially since group differences by gender and age is described in the results. What were the mean differences? Was empathy endorsement overall high? Or moderate? etc.
Response
We have added a new section and table in the results (and described it in the data analysis section of the method, lines 233-252) presenting a descriptive analysis of the variables of interest of the current study. In general, ambivalent sexism were low for girls and boys, although boys showed slightly higher mean scores in both subscales of the ASI. Regarding empathy, the descriptive statistics suggest that the scores were average for both genders, with girls showing a higher tendency to agree with the item statements than boys.
Reviewer 2 Report
Comments and Suggestions for Authors
The article Do boys empathize less than girls? Exploring the link among empathy, gender and sexist attitudes in adolescents, addresses a topic of great social relevance and contributes to the scientific advancement of a variable that seems essential in the intervention to eradicate gender violence.
Empathy is a social construct that, despite its importance, has been little studied in this context.
The paper is also well written, sets out the objectives clearly and uses the appropriate methodology to achieve the objectives.
The results, in general, provide data that deserve to be published because they lay the groundwork for further studies, so I think this article as suitable for publication in the Behavioral Sciences Journal.
Nevertheless, I consider it necessary to make some minors observations or comments that could improve the manuscript.
1) Authors presented an interesting manuscript positing the moderation effect of gender in the relation between empathy and ambivalent sexism in adolescents. The relation between empathy and ambivalent sexism is well documented in the introduction, but the role of gender in this relation is not as clear. In the present study section only appear 3 references justifying the potential moderation effect of gender, and two of these references are based on IPV offenders. Same happens in the discussion.
I think the manuscript could use other studies to fundament the role of gender on this issue and discuss it a little bit further. I leave here a couple of recent studies:
Le Brun, C., Benbouriche, M. & Tibbels, S. A (2024). Study of Empathy Towards Male Victims of Sexual Violence: The Effects of Gender and Sexism. Sexuality & Culture 28, 654–672. https://doi.org/10.1007/s12119-023-10138-3
Villanueva-Blasco et al. (2024). Teen dating violence: predictive role of sexism and the mediating role of empathy and assertiveness based on gender. Frontiers in Psychology, 15, 1393085. https://doi.org/10.3389/fpsyg.2024.1393085
These studies used some of the measures also used in the current manuscript, and their conclusions could be discussed with more detail taking into account the results of this manuscript.
2) The results section would benefit with the inclusion of one plot showing the moderation effect found in each model. Although the interaction is clear and it is well described, the visual aid provided by a graphic may facilitate its interpretation.
3) Were any missing values in the students’ responses to the questionnaires? If so, how were they handled?
Author Response
Reviewer 2
Comment
Authors presented an interesting manuscript positing the moderation effect of gender in the relation between empathy and ambivalent sexism in adolescents. The relation between empathy and ambivalent sexism is well documented in the introduction, but the role of gender in this relation is not as clear. In the present study section only appear 3 references justifying the potential moderation effect of gender, and two of these references are based on IPV offenders. Same happens in the discussion.
I think the manuscript could use other studies to fundament the role of gender on this issue and discuss it a little bit further. I leave here a couple of recent studies:
Le Brun, C., Benbouriche, M. & Tibbels, S. A (2024). Study of Empathy Towards Male Victims of Sexual Violence: The Effects of Gender and Sexism. Sexuality & Culture 28, 654–672. https://doi.org/10.1007/s12119-023-10138-3
Villanueva-Blasco et al. (2024). Teen dating violence: predictive role of sexism and the mediating role of empathy and assertiveness based on gender. Frontiers in Psychology, 15, 1393085. https://doi.org/10.3389/fpsyg.2024.1393085
These studies used some of the measures also used in the current manuscript, and their conclusions could be discussed with more detail taking into account the results of this manuscript.
Response
Thank you for highlighting this important issue and for providing relevant sources that would enhance our work. You may find these two sources included and developed in the introductory and discussion section. We hope the interaction between gender, sexism and empathy it is better explained now (lines 64-68, 92-102, 331-334).
Comment
The results section would benefit with the inclusion of one plot showing the moderation effect found in each model. Although the interaction is clear and it is well described, the visual aid provided by a graphic may facilitate its interpretation.
Response
Thank you for the suggestion. We have added a plot for each model showing the interaction to illustrate better the moderation effect (pages 7 and 9).
Comment
Were any missing values in the students’ responses to the questionnaires? If so, how were they handled?
Response
Participants were instructed to complete all items of the scales, and researchers were available during the administration of the instruments in case some of the participants have any doubt. Hence, there were no missing values. We have included this information in the procedure section (lines 203-205).
Round 2
Reviewer 1 Report
Comments and Suggestions for Authors
Thank you for the opportunity to review the revisions of this manuscript. Previously, I was primarily concerned regarding the framing and theory of the manuscript. The authors mindfully and appropriately addressed my concerns, which has strengthened the understanding of the study as a reader – and hopefully will result in a stronger contribution to the field.
One additional suggestion, however, is that in the new figures the lines for girls and boys would benefit from being more distinguishable. When printed without color, the line colors were not easily distinguishable. Perhaps using a dotted vs solid line in addition to color would increase the accessibility of the figure.
Author Response
Comment: One additional suggestion, however, is that in the new figures the lines for girls and boys would benefit from being more distinguishable. When printed without color, the line colors were not easily distinguishable. Perhaps using a dotted vs solid line in addition to color would increase the accessibility of the figure.
Response: Thank you for the suggestion. Though the instructions suggested it would be printed in colour, we have take your advice so it's clearer if there are no colours.
